# Extracellular Vesicles as Transmitters of Hypoxia Tolerance in Solid Cancers

**DOI:** 10.3390/cancers11020154

**Published:** 2019-01-29

**Authors:** Marijke I. Zonneveld, Tom G. H. Keulers, Kasper M. A. Rouschop

**Affiliations:** Maastricht Radiation Oncology (MaastRO) lab, GROW–School for Oncology and Developmental Biology, Maastricht University, 6200 MD Maastricht, The Netherlands; m.zonneveld@maastrichtuniversity.nl (M.I.Z.); tom.keulers@maastrichtuniversity.nl (T.G.H.K.)

**Keywords:** exosomes, HIF-1α, UPR, autophagy, phenocopying, preconditioning

## Abstract

Tumour hypoxia is a common feature of solid tumours that contributes to poor prognosis after treatment. This is mainly due to increased resistance of hypoxic cells to radio- and chemotherapy and the association of hypoxic cells with increased metastasis development. It is therefore not surprising that an increased hypoxic tumour fraction is associated with poor patient survival. The extent of hypoxia within a tumour is influenced by the tolerance of individual tumor cells to hypoxia, a feature that differs considerably between tumors. High numbers of hypoxic cells may, therefore, be a direct consequence of enhanced cellular capability inactivation of hypoxia tolerance mechanisms. These include HIF-1α signaling, the unfolded protein response (UPR) and autophagy to prevent hypoxia-induced cell death. Recent evidence shows hypoxia tolerance can be modulated by distant cells that have experienced episodes of hypoxia and is mediated by the systemic release of factors, such as extracellular vesicles (EV). In this review, the evidence for transfer of a hypoxia tolerance phenotype between tumour cells via EV is discussed. In particular, proteins, mRNA and microRNA enriched in EV, derived from hypoxic cells, that impact HIF-1α-, UPR-, angiogenesis- and autophagy signalling cascades are listed.

## 1. Introduction

In cancer cells, genetic and epigenetic changes allow uncontrolled growth and proliferation. In addition to these genomic alterations, the tumour microenvironment (TME) is increasingly recognized as an important contributor to cancer progression and therapy resistance [1]. The majority of solid tumours contain regions with microenvironments that are uncommon in healthy tissues. Low pH, nutrient depletion, high interstitial pressure, necrosis and hypoxia are frequently observed [2,3,4]. The high proliferative capacity and alterations in metabolism of cancer cells, as well as the highly irregular vasculature in tumours, further contribute to the existence of these features [3,5]. This results in the continuous selection of cells that have acquired resistance mechanisms to withstand these harsh conditions and contribute to increased tumour malignancy.

One of the main contributing TME features to tumour progression and malignancy is hypoxia. Tumour hypoxia is observed in the majority of solid tumours and is a very heterogeneous and dynamic feature [6]. The classical perception of tumour hypoxia is caused by a limitation of oxygen diffusion (chronic hypoxia). However, in tumours, there are additional regions displaying periodic cycling in oxygenation (acute hypoxia) [7], which can account for a large proportion of the hypoxic cells at any given time [8]. Hypoxia is therefore a very heterogeneous and dynamic feature of the TME [6]. From a clinical point of view, low oxygenation of tumours is associated with poor outcome in multiple cancer types [9], independent of treatment modality [10]. The adverse effect of tumour hypoxia is caused by the increased resistance of hypoxic cells to both chemo- and radiotherapy and the role of hypoxia-responsive mechanisms in tumour progression [11,12,13,14,15]. Additionally, there is an association between hypoxia and the occurrence of metastasis [16]. In accordance, a meta-analysis of hypoxia-modifying modalities indicated that lowering tumour hypoxia increases treatment response and patient survival [17]. To contribute to tumour regrowth after treatment or metastasis development, hypoxic cells must be reoxygenated at some time. These events of reoxygenation are important stressors on their own that contribute to the production of reactive oxygen species (ROS), activation of DNA damage responses and DNA instability [18,19].

Cells respond to hypoxia through the induction of several hypoxia tolerance mechanisms, i.e., hypoxia-inducible factor 1 α (HIF-1α) stabilization and angiogenesis, the unfolded protein response (UPR) and autophagy [15,20]. Tumour cell survival and propagation depends on the ability of tumour cells to induce these mechanisms and requires continuous communication between tumour cells and their microenvironment [21]. Collectively, these pathways alter metabolism, attenuate translation, recycle and repurpose essential building blocks to promote cellular survival and change cellular phenotype. These changes in cellular phenotype not only result in increased cell survival during acute hypoxic exposure, but also in more long-lived protection from subsequent hypoxia episodes in cells. This feature is most clearly illustrated in noncancerous tissues. 

Although prominent in the TME, during pathological conditions (i.e., stroke or infarction), normal tissues may be exposed to periods of hypoxia. In these instances, damage to normal tissue is not only sustained by cell death during the hypoxic period, but also during reperfusion by increased ROS production and inflammation [22]. Interestingly, pre-exposing (pre-conditioning) brain, heart, retina, liver and kidney tissue to hypoxia resulted in decreased cell death and reduced damage during following hypoxia episodes [23,24,25]. The pre-conditioning of cells with limited proliferative capacity indicates that cells have the capacity of reprogramming into a more hypoxia-resistant phenotype, rather than selection of cells with an intrinsic larger capacity to withstand hypoxic exposure, as often observed in cancer [26]. Moreover, the benefits of pre-conditioning are not limited to the primary hypoxic site, but can be transported to remote organs. For instance, preconditioning of limbs had beneficial effects in acute ischemic stroke (reviewed in Reference [27]), and a meta-analysis of remote pre-conditioning indicated myocardial protection [28]. These effects could even be transferred across animals through blood transfusion [29]. These and several other studies indicate systemically released factors that activate downstream hypoxia tolerance mechanisms in target cells. These include soluble factors, such as adenosine, bradykinin, opioids, nitrite/nitric oxide, stromal derived factor 1-α (SDF1-α) and calcitonin gene-related peptide (reviewed in Reference [30]), as well as extracellular vesicles (EV). In addition, an abundant population of non-membranous nanoparticles termed ‘exomeres’ was recently discovered, which can alter signaling cascades in target cells [31]. EV are important mediators of communication between neighbouring and distant cells [32]. EV consist of proteins, metabolites, lipids and genetic material, such as microRNAs (miRNA) and long noncoding RNAs (lncRNA). EV are generally considered to be a mixture of mircovesicles, exosomes and apoptotic bodies [33]. Microvesicles bud off from the plasma membrane, while exosomes are derived from fusion of the multivesicular endosome with the plasma membrane, thereby releasing its intraluminal vesicles into the extracellular space as exosomes [33]. 

Importantly, using current isolation techniques, it is not possible to distinguish between microvesicles, exosomes, and apoptotic bodies once they have been released by cells into the extracellular milieu. Therefore, this review uses the term EV to refer to all these subsets collectively. The composition of EV depends on the parental cell type, as well as on environmental factors, such as hypoxia [34]. Although EV often represent the parental cell with regard to relative abundance of content, some factors are also specifically enriched. In the case of remote preconditioning, EV-mediated transfer of miRNA resulted in profound effects on cardio protection via miR-144 [35], miR-22 [36] and miR-423-3p [37]. In addition, EV containing mRNA and protein of activating transcription factor 3 (ATF3) were shown to attenuate ischemia/reperfusion (I/R) damage to kidney cells [38,39]. Likewise, EV-associated miR-199a derived from bone marrow mesenchymal stromal cells (MSC) protected against I/R damage, potentially by suppression of UPR activation during reperfusion by targeting binding immunoglobulin protein (BiP/GR78) [40]. These studies implicate EV as important mediators for the transfer of hypoxia tolerance to remote organs. However, the underlying mechanism of remote preconditioning and how EVs mediate this remains poorly understood.

Although the phenomenon of remote preconditioning was described for normal cells, it is unclear to which extent cancer cells are capable of preconditioning/transferring hypoxia resistance to hypoxia-naïve cells. However, instances of horizontal transfer and phenocopying have been described [41,42,43]. One example is the EV-mediated transfer of drug resistance. Drug-resistant cells can incorporate functional drug-efflux pumps, such as P-glycoprotein 1 (P-gp) and ATP-binding cassette (ABC)-transporters, into EV. Upon uptake of these EV, previously drug-sensitive cells started expressing these pumps on their cell surface and were subsequently rendered resistant (reviewed in Reference [44]). Besides this direct transfer of resistance, EV also contain several miRNAs and lncRNAs that can alter cell signaling pathways, steering sensitive cells toward resistance (reviewed in References [44,45]). Sunitinib showed that renal cell carcinoma cells not only transferred resistance to other cancer cells, but also to normal endothelial cells [46], possibly contributing to therapy resistance in the entire TME. Besides therapy resistance, EV-mediated phenocopying is involved in the propagation of tumour growth and invasiveness, all contributing to increased malignancy. For instance, glioblastoma multifome (GBM) cells expressing epithelial growth factor receptor variant III (EGFRvIII), a constitutive active mutant form of EGFR, incorporate this receptor into EV. These EV can subsequently transfer it to GBM cells that did not possess the genetic mutation for vIII themselves [43], leading to increased oncogenic signaling. Similarly, in murine hepatocarcinoma, transfer of C-X-C motif chemokine receptor type 4 (CXCR4) via EV resulted in stimulation of migratory capacity [47]. In another study, Zomer et al. found that coculturing nonmigratory 4T1 cells with EV from highly migratory MDA-MB-231 cells increased the invasiveness of 4T1 cells [42]. By use of a Cre/LoxP reporter system, they were able to show that the increase in migratory capacity was only observed in 4T1 cells which had taken up EV [42]. Preliminary data of our lab and others indicate that EV derived from hypoxic cells are capable of changing the hypoxic response of cells and increase survival when exposed to low oxygen concentrations (unpublished observation, [48]).

In light of these findings, it is conceivable that hypoxic tumour cells are able to precondition surrounding cells towards a more hypoxia-tolerant phenotype through EV, leading to overall increased tumour survival and malignancy. Here, we will review the available evidence on the role of tumour hypoxia on EV cargo and its subsequent effects on the induction of the main hypoxia tolerance mechanisms in target cells, i.e., angiogenesis, HIF-1α signaling, UPR execution and autophagy.

## 2. Extracellular Vesicles & HIF1 Signaling and Angiogenesis

The most studied and best understood response to hypoxia is mediated by the HIF family of transcription factors (HIF1, HIF2, HIF3). These transcription factors consist of an instable, oxygen sensitive, α-subunit and a stable β-subunit. During normal oxygen concentrations, the instable α-subunit is rapidly hydroxylated at two proline residues in the oxygen dependent regions by one of three prolyl hydroxylases (PHDs). The α-subunit is subsequently ubiquitinated by the Von Hippel Lindau protein (VHL) and targeted for proteasomal degradation [49,50]. In the absence of oxygen, the α-subunits are not degraded and are available to dimerize with β-subunits resulting in nuclear translocation triggering a powerful transcriptional response.

HIF transcription factors regulate the expression of many genes that promote hypoxia tolerance by decreasing cellular oxygen consumption, increasing oxygen supply and regulating pH [11,51]. In general, this includes shifting the energy metabolism toward the glycolytic pathway and the expression of proteins for pH-regulation [52,53,54,55]. In addition, HIFs influence tumour oxygenation directly by promoting angiogenesis, endothelial cell survival and vasculogenesis through genes, such as vascular endothelial growth factor (VEGF), CXCR4 and SDF1 [11,52,54,55,56,57]. Hence, failure to stabilize HIF results in sensitization of cells to hypoxia and reduces the hypoxic fraction within tumours [58,59]. In addition, targeting HIF-inhibitory molecules (PHD2 and factor inhibiting HIF (FIH) through miR-182 expression) increases the degree of HIF-stabilisation during hypoxia and increases the angiogenic potential of cells [60,61]. 

One of the main reasons that the level of hypoxia within tumours correlates with poor treatment outcome is due to increased efficiency in inducing angiogenesis in support of tumour growth. Interestingly, hypoxic tumour cells induce angiogenesis through the secretion of EV ([62], unpublished observation). EV derived from hypoxic GBM cells are enriched in mRNA from HIF-1α target genes, including Bcl-2 interacting protein 3 (BNIP3), adrenamedullin (ADM), lysyl oxidase (LOX), N-myc downstream regulated 1 (NDRG1), procollagen-lysine,2-oxoglutarate 5-dioxygenase 2 (PLOD2) and serpin family E member 1 (SERPINE1) [62]. In addition to mRNA, levels of several HIF-1α-inducible proteins were enriched in these EV, including interleukin 8 (IL-8), insulin-like growth factor binding protein 1 (IGFBP1), IGFBP3 and carbonic anhydrase 9 (CAIX) [62,63]. These mRNAs and proteins are instrumental in hypoxia tolerance and their incorporation into EV suggests that an active HIF-1α phenotype can be transmitted to surrounding cells. This is further supported by findings that MSC overexpression of HIF-1α incorporated more JAGGED-1 into EV, a Notch ligand which increases angiogenesis [64]. Aga et al. even found functional HIF-1α in EV of nasopharyngeal carcinoma cells [65]. However, it is difficult to determine what the functional consequences of EV-associated HIF-1α would be, as recipient cells which experience hypoxia already express HIF-1α, while nonhypoxic cells would likely rapidly ubiquitinate and degrade the protein. Additional evidence for increased angiogenic potential of hypoxia-derived EV shows that exposure of tissue factor (TF)/VIIa on the outer membrane leaflet of EV can bind proteinase-activated receptor 2 (PAR-2) on endothelial cells leading to pro-angiogenic ERK1/2 signaling [66]. Likewise, EV-associated CAIX significantly increased angiogenesis and endothelial migration [63]. Hypoxic colorectal cell-derived EV stimulated angiogenesis through the transfer of WNT4 mRNA. This led to an upregulation of Wnt4 protein and induced β-catenin signaling in endothelial cells [67]. 

In addition to protein and mRNA, several miRNAs under the transcriptional control of HIF-1α are enriched in EV derived from hypoxic cells. Amongst these are miR-21, miR-23a, miR-126, miR-135b, miR-210. The incorporation of these hypoxia-activated miRNAs reflect the hypoxic status of the producing cells and may result in hypoxia-phenocopying in neighbouring and distant cells. In Table 1 we have summarized which miRNAs are enriched in EV derived from hypoxic cells and their effects on HIF-1α signaling and angiogenesis [68,69,70,71]. For instance, miR-21 is dependent on HIF-1α for transcription and is associated with increased resistance to apoptosis, angiogenesis and migration through the activation of Akt and ERK [72,73,74]. Its incorporation into EV supports the hypothesis that cells are able to communicate a hypoxia tolerance phenotype to other surrounding cells possibly contributing to survival of the entire tumour. Likewise, miR-23a was increased in EV from hypoxic lung cancer cells and targets PHD1 and PHD2 in acceptor cells, leading to increased HIF-1α stabilization [75]. Additionally, miR-23a targets several components necessary for apoptosis, such as BH3 interacting domain death agonist (BID), caspase-7 and NIX/BNIP3L [76], possibly heightening the threshold for hypoxia-induced cell death. Hypoxia-dependent enrichment of EV-associated miR-210 is transcriptionally regulated by HIF-1α, although many of its (predicted) target genes are not directly involved in the hypoxia response. Gene ontology analysis revealed roles for miR-210 in regulation of differentiation, membrane trafficking and amino acid catabolism, indicating that miR-210 might downregulate cellular processes which are not necessary during hypoxia, and can thus be an important contributor to tumour survival [77,78]. Other HIF-1α regulated miRNAs found in EV include miR-494, miR-127 and miR-135a [79,80,81]. Delivery of miR-494 to vascular endothelial cells targets regulators of PTEN and subsequently activates of Akt/eNOS pathway [79]. In addition, miR-494 upregulates HIF-1α expression via activation of the PI3K/Akt pathway and increases hypoxia tolerance [82]. Treatment with antagomir-494 inhibits angiogenesis and reduces tumour growth in vivo [79] Likewise, miR-4530 increases angiogenesis by targeting vasohibin 1 (VASH1) [83,84], while miR-127 was protective for I/R related tissue damage [80]. On the other hand, miR-135a targets a negative regulator of HIF-1α, estrogen-related receptor alpha (ERRα) [81]. These findings indicate that the net effect of EV will depend on the balance in the recipient cell.

## 3. Extracellular Vesicles & the Unfolded Protein Response and Autophagy

The functionality of many proteins depends on correct folding and post-translational modification. The disulfide bonds introduced during post-translational folding or isomerization are oxygen-dependent [85]. Hypoxia, therefore, results in the accumulation of misfolded proteins, leads to ER stress and rapid activation of the unfolded protein response (UPR) [86], allowing cells to survive hypoxia exposure [12,58,87,88]. This is an evolutionarily conserved pathway that responds to endoplasmic reticulum (ER) stress by the coordinate action of three ER stress sensors present within the ER membrane, protein kinase-like ER kinase (PERK/EIF2AK3), inositol-required enzyme 1 (IRE1/ERN1) and ATF6 [89]. In the absence of ER stress, these sensors are associated with BiP/GRP78, preventing their activation. BiP is a chaperone protein involved in proper protein assembly. During ER stress, it is sequestered by misfolded proteins, leading to activation of the three transmembrane proteins and their respective UPR branches in order to restore ER homeostasis. The balance between BiP expression and UPR-executors (i.e., the quantity of “free” BiP) determines the threshold and extent of UPR-activation. Despite several reports which show that ER stress and UPR can be transported to other surrounding cells [75,90], EV as vehicles of this transport have mostly been ignored. However, recently, tumour EV were shown to induce the IRE1 branch of the UPR in nonmalignant target cells [22]. Although this was not studied in the context of hypoxia, it suggests that an increased hypoxia-tolerant phenotype could be transmitted by EV through activation of UPR. However, indirect evidence for the involvement of EV exists, as some miRNAs that have been shown to be associated to EV from hypoxic tumour cells in other studies are known to influence UPR signaling (Table 1). For example, miR-204 which targets PERK [91] or miR-181a. MiR-181a is associated with decreased BiP/GRP78 protein levels, but increased levels of mRNA, suggesting translational arrest [92]. By reducing or blocking miR-181a the brain was protected from stroke [93]. MiR-433 reduces glutathione biosynthesis, in turn leading to more oxidative stress and UPR activation [94].

In addition to HIF-1α and UPR, hypoxia activates autophagy, the lysosomal degradation pathway that mediates both selective and bulk degradation of proteins, cytoplasmic content and organelles [12,95,96,97,98,99,100]. Nonselective autophagy is referred to as macro-autophagy (autophagy in this review), whereas micro-autophagy, mitophagy and chaperone-assisted autophagy refer to more selective forms that are directed by specific proteins. Autophagy begins with the formation of a double-membrane structure, the autophagosome, to engulf cellular content. Once formed, the outer membrane fuses with a lysosome, releasing its contents to the degradative enzymes. Autophagy plays an important role during conditions of starvation or metabolic stress by ‘recycling’ amino acids and nutrients to maintain energy levels, protein synthesis and essential metabolic processes [98]. Failure to execute autophagy in response to hypoxia results in cell death and sensitization of tumours to therapy [12,96]. Furthermore, autophagy is essential for removal of mitochondria that might become cytotoxic and is used during cycling hypoxia to rapidly decrease mitochondrial mass and reduce ROS production [100,101]. As a consequence, maintaining high levels of autophagy during hypoxia is essential for survival and inhibiting autophagy exposes the cells to increased ROS and cell death [12,102,103]. Hypoxic cells can secrete various signaling factors including proteins and/or microRNAs that are capable of influencing the autophagic potential of distant cells and potentially prime the cells to withstand periods of low oxygenation through elevated autophagy induction.

Increased mRNA levels of the HIF-1α transcriptional target, BNIP3, are observed in EV derived from hypoxic GBM cells [62]. BNIP3 induces a type of cell death that has features of necrosis, rather than apoptosis, and which was associated with the production of autophagic bodies [104], later described as programmed cell death II [105]. Most likely, BNIP3 influences autophagy through its interaction with B-cell CLL/lymphoma 2 (Bcl-2). Bcl-2, in addition to its apoptosis regulating capacity, binds and inhibits the essential autophagy protein Beclin 1 (BECN1 is also known as ATG6) [106,107]. Several miRNA’s upregulated in EV from hypoxic cancer cells have been described to influence autophagy, although it is often unclear whether the net effect is stimulatory or inhibitory (Table 1). For instance, miR-181 was described to have autophagy-inducing effects through interaction with the Bcl-2/Beclin axis [108]. On the other hand, autophagy-inhibiting effects were also described through miR-181a targeting of myotubularin-related protein 3 (MTMR3) and autophagy-related protein 5 (ATG5) [109,110]. Likewise, MiR-125 can induce autophagy through downregulation of fork-head box P3 (FoxP3) [111], but can inhibit autophagy by targeting UV radiation resistance associated gene (UVRAG) [112,113]. MiR-23a can promote autophagy by modulating X-linked inhibitor of apoptosis (XIAP) [114], as well as inhibit it by targeting ATG3 [115]. MiR-335 targets superoxide dismutatase 2 (SOD2) [116], which triggers autophagy [117]. MiR-433 targets glutathione biosynthesis, which leads to increased oxidative stress and elevated UPR- and autophagy-activation [94]. On the other hand, hypoxic prostate cancer cells upregulate miR-204 and miR-143, which target microtubule associated protein 1 light chain β (MAP1LC3B) and ATG2B, respectively, inhibiting autophagy [118,119].

## 4. The Importance of EV Isolation Methods for the Interpretation of Results

Isolation methods for EV are known to impact findings with regard to composition and function and are, subsequently, important to note for the interpretation of results. For instance, it was shown that the vast majority of extracellular RNA is, in fact, not EV-associated, but rather coupled to RNA carrier proteins, such as Argonaut-2 [144,145,146] or high-density lipoproteins [147]. However, this extracellular RNA, as well as lipoprotein particles, soluble protein, protein aggregates, and the recently discovered exomeres are co-isolated by ultracentrifugation at 100,000 g, one of the most commonly applied isolation methods for EV [33,148,149,150]. The same holds true for polyethylene glycol (PEG) precipitation-based isolation methods, such as exoquick [146]. Although it is becoming more recognized that further purification steps, such as density gradient separation or size exclusion chromatography, are necessary to be able to distinguish between true EV effects or effects resulting from other biologically relevant sources, this has yet to become common practice [151,152,153,154]. However, these purification steps are relatively labour-intensive, require specialized equipment or result in significant loss of EV particles, hampering widespread implementation. For purposes of this review, we therefore did not discriminate based on EV isolation methods. However, as we do believe that these isolation methods are key to fully understanding published results, we summarized the EV isolation methods and biological sources used by the studies we discussed in Table 2. Where available, we also included the EV-TRACK scores of the papers discussed, as EV-TRACK aims to concisely report important methodological parameters for EV-related publications, creating more transparency and facilitating interpretation of results [153]. However, it should be taken into account that many publications are not yet recorded in the EV-TRACK database due to its recent development and the relevance of these publications cannot be evaluated based on these scores alone.

## 5. Conclusions

Overall, there is a great deal of evidence pointing toward a key role for EV in tumour hypoxia tolerance induction (proposed model in Figure 1). Besides an ability to transfer ready-to-use molecules to other cells, miRNA’s are able to finetune pathways necessary for cell survival during hypoxia. In addition, EV-mediated communication can alleviate long-term hypoxia by interacting with endothelial cells in the TME leading to angiogenesis. However, it should be noted that reports on hypoxia-derived EV that have been performed with EV purified from protein and RNA contaminants are currently lacking, and caution should be taken when interpreting these results as EV-mediated. As most extracellular RNA and protein is, in fact, not EV-associated [146], it is currently difficult to determine what the true hypoxia modulating capacity of EV is. This makes additional fundamental research, with protocols that include current knowledge on isolation procedures, into the effects of hypoxia on EV cargo and effects necessary.

## Figures and Tables

**Figure 1 cancers-11-00154-f001:**
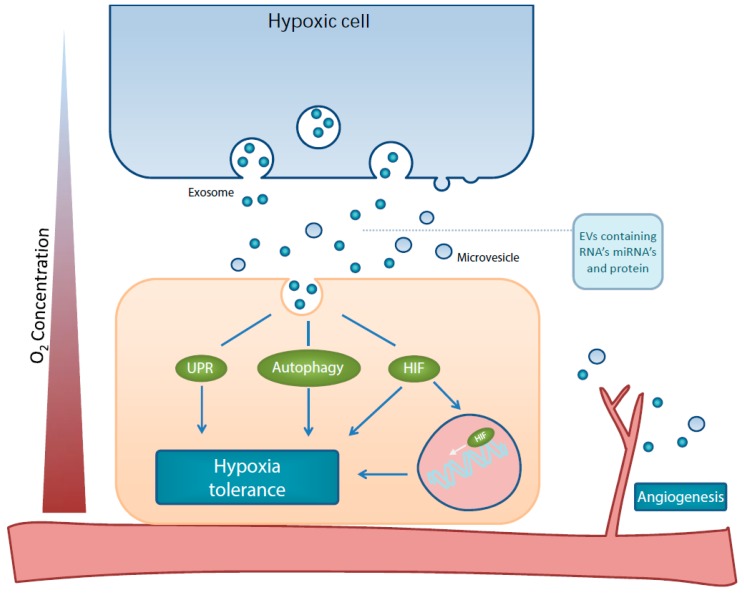
Schematic representation of effects by hypoxia-derived EVs on the hypoxia tolerance mechanisms, HIF, UPR, autophagy and angiogenesis.

**Table 1 cancers-11-00154-t001:** Summary of miRNAs found to be upregulated in EV from hypoxic tumours and their possible effects on HIF-1α, UPR and autophagy.

miRNA	HIF-1α		UPR		Autophagy	
	Main Effect	Reference	Main Effect	Reference	Main Effect	Reference
miR-21 [69]	miR-21 induces tumour angiogenesis through targeting PTEN, leading to activate AKT and ERK1/2 signaling pathways, and thereby enhancing HIF-1α	[73,74]	unknown		Targets Rab11a inhibiting autophagy	[120]
miR-23a [71]	Targets PHD1 and PHD2 leading to HIF-1α stabilization	[71]	Reduces UPR activation	[121]	Elevated mir23a induces autophagy though XIAP mediated autophagy	[114]
					miR-23a inhibits autophagy by targeting ATG3	[115]
miR-92a [68]	Targets VHL	[122]	unknown		unknown	
miR-125 [69]	negatively regulated by HIF1; induces mitochondrial fission	[123]	degraded by IRE1; targets caspase-2	[124]	inhibits autophagy activation by targeting UVRAG	[112]
	inhibits angiogenesis	[125]			activates autophagy by targeting FOXP3	[111]
niR-127 [68]	Protective against I/R damage; Under transcriptional control of HIF-1α	[80]	unknown		unknown	
miR-135a [70]	expression is HIF-dependent	[81]	unknown		ATG14 is a target gene	[126]
	Targets ERRα. ERRα augments HIF1, so downregulates HIF signaling	[127,128]				
miR-143 [68]	unknown		unknown		Inhibits ATG2B and thus autophagy	[119]
miR-181 [68,69]	enhances VEGF expression	[129]	Regulates BiP/GRP78	[92]	decreases autophagy by regulating the p38 MAPK/JNK pathway	[130]
	increases angiogenesis by targeting PDCD10 and GATA6	[131]				
miR-204 [68]	unknown		Targets PERK	[91]	Suppresses tumour growth; targets LC3B	[118]
					Targets Bcl-2 an inhibitor of autophagy	[132]
					Targets TRPM3 a stimulator of autophagy	[133]
miR-292 [68]	unknown		unknown		Targets ATG7 and ULK1	[134]
miR-335 [68]	Inverse expression profile to HIF1a	[120]	unknown		inhibits SOD2, which triggers autophagy	[116,117]
					Targets FASN, which stimulates AMPK/ULK1	[135]
miR-433 [68]	targets HIF-1α	[136]	Reduces glutathione biosynthesis leading to more oxidative stress	[94]	Reduces glutathione biosynthesis leading to more oxidative stress	[137]
miR-451 [68]	unknown		unknown		inhibits TSC1 stimulating autophagy	[138]
miR-494 [70]	unknown		unknown		Increases autophagy (but not conclusive, ratio not flux)	[139]
miR-513a [70]	unknown		unknown		unknown	
miR-542 [68]	unknown		unknown		induced PI3K/Akt signaling	[134,140]
miR-547a [68]	unknown		unknown		unknown	
miR-575 [70]	unknown		unknown		unknown	
miR-885 [68]	unknown		unknown		Targets MDM4, ATK1, BCL2, ATG16L2, ULK2, CASP2, and CASP3	[141]
miR-1233-1 [70]	unknown		unknown		unknown	
miR-4463 [70]	unknown		unknown		Inhibits XIAP and Bcl-2. This can lead to enhanced autophagy	[142]
miR-4497 [70]	unknown		unknown		unknown	
miR-4498 [70]	unknown		unknown		unknown	
miR-4530 [70]	Increases angiogenesis. VASH1 is a target gene	[83]	unknown		regulates autophagy by targeting SIRT1 and activating PI3K/AKT/mTOR	[143]
miR-4721 [70]	unknown		unknown		unknown	
miR-4728 [70]	unknown		unknown		unknown	
miR-4741 [70]	unknown		unknown		unknown	
miR-4763 [70]	unknown		unknown		unknown	
miR-6087 [70]	unknown		unknown		unknown	
miR-6132 [70]	unknown		unknown		unknown	

**Table 2 cancers-11-00154-t002:** Isolation methods and EV-METRIC scores used by the studies discussed. EV-TRACK ID as referenced in the platform for Transparant Reporting And Centralizing Knowledge in Extracellular Vesicles research (EV-TRACK [153]). EV-METRIC score represents the completeness of reporting of generic and method-specific information necessary to interpret and reproduce the experiment. N/A: Not available.

Reference	Main Finding	EV Isolation Method	EV Source	EV-Track ID [153]	EV-METRIC (%)
[66]	TF/VIIa on EV activate endothelial ERK1/2	Ultracentrifugation	Cell culture supernatant	EV110023	33
[35]	EV-mediated cardioprotection after remote ischemic preconditioning	Exoquick	Murine serum	N/A	-
[36]	EV-mediated cardioprotection after remote ischemic preconditioning	Exoquick and ultracentrifugation	Cell culture supernatant	EV140155	13
[42]	EV from migratory cells increased migration of non-migratory cells	N/A (in vivo monitoring)	N/A	N/A	-
[43]	EGFRvIII is transferred between cells via EV	Ultracentrifugation	Cell culture supernatant & Murine plasma	N/A	-
[37]	EV-mediated cardioprotection after Remote ischemic preconditioning	Exoquick	Cell culture supernatant	N/A	-
[38]	EV attenuates inflammation after renal I/R	Ultracentrifugation	Human/murine urine & cell culture supernatant	EV140313	0
[39]	ATF3 is present in urine EV	Ultracentrifugation	Human urine	N/A	-
[41]	EV from melanoma cells with different metastatic potential contain distinct proteins and RNA’s	Ultracentrifugation	In vivo grown tumour tissue	N/A	-
[46]	EV associated lncARSR causes sunitinib resistance	Ultracentrifugation	Cell culture supernatant	N/A	-
[47]	Transfer of CXCR4 via EV	Exoquick	Cell culture supernatant	N/A	-
[155]	EV from hypoxic prostate cancer cells target adherens junctions in hypoxia naïve cells	Exoquick and ultracentrifugation	Cell culture supernatant	EV140124	25–43
[156]	HIF-1α and RAB22A stimulate metastases promoting EV secretion	High speed centrifugation	Cell culture supernatant	EV140412	0
[157]	Hypoxia promotes EV release via HIF-1α	Exoquick and ultracentrifugation	Cell culture supernatant	EV120021	25–33
[62]	EV from hypoxic cells resemble parental cell	Ultracentrifugation	Cell culture supernatant, murine plasma, human plasma	EV130043	33
[63]	CAIX on EV increased angiogenesis and endothelial migration	Ultracentrifugation	Cell culture supernatant	N/A	-
[64]	EV stimulate angiogenesis through JAGGED-1	Ultracentrifugation	Cell culture supernatant	N/A	-
[65]	HIF-1α is transferred via EV	Ultracentrifugation	Cell culture supernatant	EV140293	11
[67]	EV-mediated transfer of WNT4 mRNA	Total exosome isolation kit	Cell culture supernatant	N/A	-
[68]	miRNA profile of EV from hypoxic prostate cancer cells	Ultracentrifugation and/or Exoquick; unclear	Cell culture supernatant	N/A	-
[69]	miRNA profile of EV from hypoxic epithelial ovarian cancer cells	Total exosome isolation kit	Cell culture supernatant	N/A	-
[70]	miRNA profile of EV from hypoxic melanoma cells	Ultracentrifugation	Cell culture supernatant	N/A	-
[71]	Hypoxic lung cancer cells secrete EV with miR-23a, increasing HIF-1α stabilization in target cells	Total exosome isolation kit	Cell culture supernatant & human serum	N/A	-

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
