# Peer review of "Extracellular Vesicles as Transmitters of Hypoxia Tolerance in Solid Cancers"

_cancers, 2019, doi:10.3390/cancers11020154_

Round 1

Reviewer 1 Report

The manuscript discusses the role of extracellular vesicles in the transmission of tolerance to hypoxia, an emerging and mechanistically important field of tumorigenesis. The review is, in general, concise and well written and provides an overview on the role of a list of microRNAs upregulated in hypoxic tumours and their possible role in hypoxia tolerance. Publications cited in the review are weighted by giving information of the completeness of information on the extracellular vesicles' preparation. This type of information is of increasing importance in view of comparison of results and the generation of protocols approved for clinical application.

I suggest acceptance of the review after minor revision as stated below:

Lines 16-17: "The extent of hypoxia...": It's rather the damage due to hypoxia than the extent of hypoxia that is determined by the tumour cells' capability to activate hypoxia-tolerance mechanisms.

43: "periodic cycling in oxygenation (acute hypoxia)": Reperfusion (as also mentioned later in the manuscript) is at least as big of a problem as hypoxia per se. Please shortly state the role reperfusion.

81-84: The role of apoptotic bodies is not described.

89: ATF3: First appearance of abbreviation, needs to be spelled out. There are many other abbreviations, mostly of genes/proteins, that are not spelled out throughout the article. I leave it to the editors to decide on the type of abbreviations that should be spelled out.

107: The term GBM cells, other than names of genes and proteins, should be spelled out at its first appearance, in any case.

120: "towards a more hypoxic phenotype": should it rather read hypoxia-tolerant?

141-142: "results" vs "reduced": use the same tense within a sentence.

154: "mesenchymal stem cells": It is now generally accepted that the term "mesenchymal stromal cells" should be used instead.

162: The authors should state what the consequences of ERK1/2 activation are in the given context.

183: "roles for... in regulation of" per se gives no information on up- or down-regulation; please be more specific.

186: "Mir-494": how is the exception in the nomenclature justified (according to the species studied in the cited publication?)

192: "I/R": please spell out (first occurrence).

202: The general reader of the manuscript would profit from a description of the reasons for misfolded proteins in ER stress.

207ff: A quick PubMed search revealed several recent articles that should be cited:

Wu CH, Silvers CR, Messing EM, Lee YF. Bladder cancer extracellular vesicles drive tumorigenesis by inducing the unfolded protein response in endoplasmic reticulum of non-malignant cells. J Biol Chem. 2018 Dec 28. pii: jbc.RA118.006682. doi: 10.1074/jbc.RA118.006682.

Wang C, Zhu G, He W, Yin H, Lin F, Gou X, Li X.BMSCs protect against renal ischemia-reperfusion injury by secreting exosomes loaded with miR-199a-5p that target BIP to inhibit endoplasmic reticulum stress at the very early reperfusion stages. FASEB J. 2019 Jan 14:fj201801821R. doi: 10.1096/fj.201801821R.

246: "induced autophagy through MTMR3 [103] and ATG5 [104].": Please be more specific (inhibition of, targeting, downregulation etc.).

Table 1: Consider writing "unknown" instead of "none" if there is no information available to distinguish from no effect.

Author Response

Dear editor and referee,

Please find below an itemized response to the issues that were raised based on the review of our manuscript. For clarity, changes in the main body of the manuscript have been marked with track changes. Thank you very much for your criticisms. We have tried to address them in the best way we could, and we believe that the manuscript has improved significantly.

Lines 16-17: "The extent of hypoxia...": It's rather the damage due to hypoxia than the extent of hypoxia that is determined by the tumour cells' capability to activate hypoxia-tolerance mechanisms.

The point raised here, is that increased hypoxic fraction is (in part) dependent on the capability of tumour cells to survive hypoxia. Low hypoxia tolerance will lead to killing of hypoxic cells and low hypoxic fraction and vice versa. To clarify this, we changed the text into:

The extent of hypoxia within a tumour is influenced by the tolerance of individual tumor cells to hypoxia, a feature that differs considerably between tumors. High numbers of hypoxic cells may therefore be a direct consequence of enhanced cellular capability in activation of hypoxia-tolerance mechanisms.”

43: "periodic cycling in oxygenation (acute hypoxia)": Reperfusion (as also mentioned later in the manuscript) is at least as big of a problem as hypoxia per se. Please shortly state the role reperfusion.

We agree with the reviewer and have shortly added the role of reoxygenation in acute hypoxia.

Line54-58: “To contribute to tumour regrowth after treatment or metastasis development, hypoxic cells must at some time be reoxygenated. These events of reoxygenation are important stressors on their own that contribute to the production of reactive oxygen species (ROS), activation of DNA damage responses and DNA-instability [18,19]

And line 70-73:

Although prominent in the TME, during pathological conditions (i.e. stroke or infarction), normal tissues may be exposed to periods of hypoxia. In these instances, damage to normal tissue is not only sustained by cell death during the hypoxic exposure, but also during reperfusion by increased (ROS) production and inflammation [22].

 81-84: The role of apoptotic bodies is not described.

There are (so far) no clear indications that apoptotic bodies are involved in hypoxia tolerance mechanisms. These are therefore not explicitly mentioned in our review. However, the EV isolation methods used in the studies discussed will most likely co-isolate apoptotic bodies. To clarify this, we have added lines 89-91 to our text.

Importantly, it is currently not possible to distinguish between microvesicles, exosomes, exomeres and apoptotic bodies once they have been released by cells into the extracellular milieu. Therefore, this review uses the term EV to refer to all these subsets collectively.”

89: ATF3: First appearance of abbreviation, needs to be spelled out. There are many other abbreviations, mostly of genes/proteins, that are not spelled out throughout the article. I leave it to the editors to decide on the type of abbreviations that should be spelled out.

We thank the reviewer for their astute observation and have spelled out all first time abbreviations in the text.

107: The term GBM cells, other than names of genes and proteins, should be spelled out at its first appearance, in any case.

This has been adjusted accordingly.

120: "towards a more hypoxic phenotype": should it rather read hypoxia-tolerant?

We agree with the reviewer and have adjusted the text accordingly.

141-142: "results" vs "reduced": use the same tense within a sentence.

This has been adjusted accordingly

154: "mesenchymal stem cells": It is now generally accepted that the term "mesenchymal stromal cells" should be used instead.

We have adjusted the terminology from ‘mesenchymal stem cells’ to ‘’mesenchymal stromal cells’ during the first use of the term in line 99 of the revised manuscript.

162: The authors should state what the consequences of ERK1/2 activation are in the given context.

We have revised the manuscript to read: ‘Additional evidence for increased angiogenic potential of hypoxia-derived EV shows that exposure of TF/VIIa on the outermembrane leaflet of EV can bind PAR-2 on endothelial cells leading to pro-angiogenic ERK1/2 signalling activation [63]’. Lines 172-175 of the revised manuscript.

183: "roles for... in regulation of" per se gives no information on up- or down-regulation; please be more specific.

We have purposely not reported whether miR-210 up- or down-regulates differentiation, membrane trafficking or amino acid catabolism, as the nett effect is hypothesized to be the downregulation of processes unnecessary during hypoxia, which will be cell specific.

186: "Mir-494": how is the exception in the nomenclature justified (according to the species studied in the cited publication?)

The difference in nomenclature arose from a typo. We apologize to the reviewer and have adjusted the text accordingly.

192: "I/R": please spell out (first occurrence).

We thank the reviewer for their comment and have spelled out the first occurrence in line 98 of the revised manuscript.

202: The general reader of the manuscript would profit from a description of the reasons for misfolded proteins in ER stress.

We have adjusted the text to include a brief description in the revised manuscript.

The functionality of many proteins depends on correct folding and post-translational modification. The disulfide bonds introduced during post-translational folding or isomerization are oxygen dependent{Koritzinsky, 2013 #1616}. Hypoxia therefore results in the accumulation of misfolded proteins, leads to ER stress and rapid activation of the unfolded protein response (UPR) [83] allowing cells to survive hypoxia exposure [12,57,84,85]

207ff: A quick PubMed search revealed several recent articles that should be cited:

Wu CH, Silvers CR, Messing EM, Lee YF. Bladder cancer extracellular vesicles drive tumorigenesis by inducing the unfolded protein response in endoplasmic reticulum of non-malignant cells. J Biol Chem. 2018 Dec 28. pii: jbc.RA118.006682. doi: 10.1074/jbc.RA118.006682.

Wang C, Zhu G, He W, Yin H, Lin F, Gou X, Li X.BMSCs protect against renal ischemia-reperfusion injury by secreting exosomes loaded with miR-199a-5p that target BIP to inhibit endoplasmic reticulum stress at the very early reperfusion stages. FASEB J. 2019 Jan 14:fj201801821R. doi: 10.1096/fj.201801821R.

We thank the reviewer for pointing out these recent publications and have included them in the revised manuscript (reference 22 & 39)

However, recently, tumour EV were shown to induce the IRE1 branch of the UPR in non-malignant target cells [22]. Although this was not studied in the context of hypoxia, it suggests that an increased hypoxia tolerant phenotype could be transmitted by EV through activation of UPR.”

“Likewise, EV-associated miR-199a derived from bone marrow mesenchymal stromal cells (MSC) protected against I/R damage, potentially by supression of UPR activation during reperfusion by targeting binding immunoglobulin protein (BiP/GR78)[39].”

246: "induced autophagy through MTMR3 [103] and ATG5 [104].": Please be more specific (inhibition of, targeting, downregulation etc.).

As suggested, this section was rewritten into a more clear format on inhibitory and stimulating actions.

Several miRNA’s upregulated in EV from hypoxic cancer cells have been described to influence autophagy, although it is often unclear whether the net effect is stimulatory or inhibitory (Table 1). For instance, miR -181 has been described to have autophagy inducing effects through interaction with the Bcl-2/Beclin axis [106]. On the other hand, autophagy inhibiting effects have also been described through miR-181a targeting of myotubularin-related protein 3 (MTMR3) and autophagy-related protein 5 (ATG5)  [107] [108]. Likewise, MiR-125 can induce autophagy through downregulation of forkhead box P3 (FoxP3) [109], but can inhibit autophagy by targeting UV radiation resistance associated gene (UVRAG) [110,111]. MiR-23a can promote autophagy by modulating X-linked inhibitor of apoptosis (XIAP) [112], as well as inhibit it by targeting ATG3 [113]. MiR-335 targets superoxide dismutatase 2 (SOD2) [114], which triggers autophagy [115].”

Table 1: Consider writing "unknown" instead of "none" if there is no information available to distinguish from no effect.

We agree with the reviewer and have altered all occurrences of ‘none’ to ‘unknown’ in Table 1.

Reviewer 2 Report

The review “Extracellular Vesicles as transmitters of hypoxia tolerance in solid cancers” addresses an interesting issue that deserves to be addressed in a review in a objective way as successfully done by Zonneveld & co. I liked reading the manuscript and have no major concerns. I would still want to suggest a couple of improvements:

-The review would benefit from a figure that could present a model about the main effects of EVs related to hypoxia tolerance based on the current knowledge presented in the text

-The authors could also elaborate, whether the publications discussed include data from cell cultures, animal models or patients – this affects the validity of the data as much as the EV isolation method

-line 79, it would be good to add that EVs contain metabolites in addition to proteins, lipids and genetic material and line 81-82: it could be said that additional types of EVs are being discovered (such as exomeres)

-line 125 (title): remove parenthesis i.e. decide whether angiogenesis should be there or not

-lines 245-249 data not well explained, elaborate more

-lines 258-260 and later ref 114 has a lot of weight, but other (conflicting) publications on the subject also exists. it would be good to add references to those as well.

-EV purification technologies, lines 265-266, would be better to mention that density gradients and size-exclusion chromatography are not challenge-free technologies either (being laborious or suffer from loss of EVs and still some contaminants)

-EV Track is relatively new, so the text should mention that publications should not be evaluated based on the EV track score only

-check that abbreviations are spelled out in a consistent way

-check the language (commas, articles, prepositions, sentence structures etc): lines 57, 129, 149, 164, 188, 196, 200, 210-212, 252-255, 263

Author Response

Dear editor and referee,

Please find below an itemized response to the issues that were raised based on the review of our manuscript. For clarity, changes in the main body of the manuscript have been marked with track changes. Thank you very much for your criticisms. We have tried to address them in the best way we could, and we believe that the manuscript has improved significantly.

-The review would benefit from a figure that could present a model about the main effects of EVs related to hypoxia tolerance based on the current knowledge presented in the text

We have added Figure 1 that summarizes the main effects of hypoxia-derived EVs on target cells to the revised manuscript as a proposed model.

-The authors could also elaborate, whether the publications discussed include data from cell cultures, animal models or patients – this affects the validity of the data as much as the EV isolation method

We agree with the reviewer that adding the source of EV would be of added value and have added an extra column to Table 2 that describes the source of the EVs.

-line 79, it would be good to add that EVs contain metabolites in addition to proteins, lipids and genetic material and line 81-82: it could be said that additional types of EVs are being discovered (such as exomeres)

We have added ‘metabolites’ to the description of EV contents in line 84 of the revised manuscript. We have also adjusted the text to clarify better what we consider to be EV in this review.

EV consist of proteins, metabolites, lipids, and genetic material, such as microRNA’s (miRNA) and long non-coding RNA’s (lncRNA). EV are generally considered to be a mixture of mircovesicles, exosomes, and apoptotic bodies [32]. Microvesicles bud off from the plasma membrane, while exosomes are derived from fusion of the multivesicular endosome with the plasma membrane, thereby releasing its intraluminal vesicles into the extracellular space as exosomes [32]. Recently,  an abundant population of non-membranous nanoparticles termed 'exomeres' has been discovered that can alter signaling cascades in target cells[33].  Importantly, using current isolation techniques, it is not possible to distinguish between microvesicles, exosomes, exomeres and apoptotic bodies once they have been released by cells into the extracellular milieu. Therefore, this review uses the term EV to refer to all these subsets collectively.”

-line 125 (title): remove parenthesis i.e. decide whether angiogenesis should be there or not

We thank the reviewer for this observation and have removed the parenthesis as suggested.

-lines 245-249 data not well explained, elaborate more

We have adjusted the text to more clearly explain the data.

Several miRNA’s upregulated in EV from hypoxic cancer cells have been described to influence autophagy, although it is often unclear whether the net effect is stimulatory or inhibitory (Table 1). For instance, miR -181 has been described to have autophagy inducing effects through interaction with the Bcl-2/Beclin axis [108]. On the other hand, autophagy inhibiting effects have also been described through miR-181a targeting of myotubularin-related protein 3 (MTMR3) and autophagy-related protein 5 (ATG5)  [109] [110]. Likewise, MiR-125 can induce autophagy through downregulation of forkhead box P3 (FoxP3) [111], but can inhibit autophagy by targeting UV radiation resistance associated gene (UVRAG) [112,113]. MiR-23a can promote autophagy by modulating X-linked inhibitor of apoptosis (XIAP) [114], as well as inhibit it by targeting ATG3 [115]. MiR-335 targets superoxide dismutatase 2 (SOD2) [116], which triggers autophagy [117]. MiR-433 targets glutathione biosynthesis, which leads to increased oxidative stress and elevated UPR- and autophagy-activation [94].”

-lines 258-260 and later ref 114 has a lot of weight, but other (conflicting) publications on the subject also exists. it would be good to add references to those as well.

We have added additional references and included other possible RNA carriers to the text (lines 286 – 287).

-EV purification technologies, lines 265-266, would be better to mention that density gradients and size-exclusion chromatography are not challenge-free technologies either (being laborious or suffer from loss of EVs and still some contaminants)

We agree with the reviewer and have adjusted the text accordingly in the revised manuscript.

Although it is becoming more recognized that further purification steps, such as density gradient separation or size exclusion chromatography are necessary to be able to distinguish between true EV effects or effects resulting from other biologically relevant sources, this has yet to become common practice [127-130]. However, these purification steps are relatively labour intensive, require specialized equipment or result in significant loss of EV particles, hampering widespread implementation.”

-EV Track is relatively new, so the text should mention that publications should not be evaluated based on the EV track score only

We agree with the reviewer and have adjusted the text accordingly

However, it should be taken into account that many publications are not yet recorded in the EV-TRACK database due to its recent development and the relevance of these publications cannot be evaluated based on these scores alone.”

-check that abbreviations are spelled out in a consistent way

We apologize to the reviewer and have adjusted the text throughout.

-check the language (commas, articles, prepositions, sentence structures etc): lines 57, 129, 149, 164, 188, 196, 200, 210-212, 252-255, 263

We apologized to the reviewer and have adjusted the language in the lines mentioned, and checked the manuscript throughout.